# Application of Distributed Acoustic Sensing for Active Near-Surface Seismic Monitoring

**DOI:** 10.3390/s25051558

**Published:** 2025-03-03

**Authors:** Eslam Roshdy, Mariusz Majdański, Szymon Długosz, Artur Marciniak, Paweł Popielski

**Affiliations:** 1Institute of Geophysics, Polish Academy of Sciences, 01-452 Warsaw, Poland; emohamed@igf.edu.plamarciniak@igf.edu.pl (A.M.); 2Geophysics Department, Faculty of Science, Cairo University, Giza 12613, Egypt; 3SHM System/Nerve-Sensors, 30-444 Libertów, Poland; szymon.dlugosz@shmsystem.pl; 4Faculty of Building Services, Hydro and Environmental Engineering, Warsaw University of Technology, 00-661 Warszawa, Poland; pawel.popielski@pw.edu.pl

**Keywords:** distributed acoustic sensing, 3C geophone, near-surface, seismic monitoring

## Abstract

High-resolution imaging of the near-surface structures of critical objects is necessary in various applications including geohazard studies, the structural health of artificial structures, and generally in environmental seismology. This study explores the use of fiber optic sensor technology in active seismic surveys to monitor the embankment structure of the Rybnik Reservoir in Poland. We discuss the technical aspects, including sensor types and energy sources, and provide a comparison of the data collected with a standard geophone-based survey conducted simultaneously. A thorough data processing methodology is presented to directly compare both datasets. The results show a comparable data quality, with DAS offering significant advantages in terms of both the spatial and temporal resolution, facilitating more accurate interpretations. DAS demonstrates its ability to operate effectively in complex geological environments, such as areas with high seismic noise, rough terrain, and variable surface conditions, making it highly adaptable for monitoring critical infrastructure. Additionally, DAS provides long-term monitoring capabilities, essential for ongoing structural health assessments and geohazard detection. For example, the multichannel analysis of surface waves (MASW) using DAS data clearly identifies S-wave velocities down to 13 m with an RMS error of 3.26%, compared to an RMS error of 6.2% for geophone data. Moreover, the DAS-based data were easier to process and interpret. The integration of DAS with traditional seismic data can provide a more comprehensive understanding of subsurface properties, facilitating more accurate and reliable geophysical assessments over time. This innovative approach is particularly valuable in challenging environments, underscoring its importance in monitoring critical infrastructure.

## 1. Introduction

Knowledge about the near-surface environment is critical for understanding geohazards, structural health, and hydrogeological processes. Standard active seismic, geophone-based surveys can lead to clean, high-resolution images of the difficult structures [1,2] and enable the tracking of environmental processes with time-lapse observations [3]. However, the effective seismic monitoring of critical infrastructure faces significant challenges due to its highly heterogeneous and dynamic nature. Geophone arrays often lack the spatial and temporal resolution required to capture subtle changes in seismic wave propagation, limiting their effectiveness for applications such as structural health monitoring, geohazard detection, and environmental seismology [4].

Over the past few years, Distributed Fiber Optic Sensing (DFOS) has received significant attention across various technical fields. Optical fiber sensors offer robust, accurate, and reliable measurements of axial deformations in the fiber, while maintaining relatively low installation costs and a high resistance to hazardous environmental conditions. In seismic applications, the most widely used DFOS system is Distributed Acoustic Sensing (DAS) [5].

DAS offers the capability of measuring thousands of spatial points simultaneously, using a simple, distributed, and unmodified optical fiber as the sensing element to measure strain primarily. It is defined as the relative change in the fiber length over a specific segment, normalized by the gauge length. DAS can also provide insights into temperature changes and fractures, also useful in environmental monitoring. This technology detects phase changes in backscattered light, which are directly proportional to the strain, enabling the detection of strain rates as small as nanometers per second with sampling frequencies in the tens of kilohertz [6]. This capability, along with its high sensitivity and resolution, allows DAS to capture acoustic signals effectively, giving rise to its name “Acoustic”. Unlike conventional point sensors, DAS monitors the entire fiber length by dividing it into virtual sections based on the gauge length. Measurements are recorded at the midpoint of each section, with a Gaussian window applied to match the width of the laser pulse used during interrogation. The channel spacing can reach 20 cm, offering an excellent spatial resolution not possible using standard geophones [7].

Additionally, DAS systems cover a broad frequency range even up to 20 kHz, making them suitable for diverse seismic applications despite their unidirectional data output. One of DAS’s key strengths lies in its adaptability. Parameters such as the gauge length and channel spacing can be easily modified, enabling a single optical fiber to conduct both deep and shallow surveys simultaneously. This versatility makes DAS a powerful and efficient tool for seismic monitoring [8]. This ultra-high-resolution data acquisition opens new frontiers for near-surface seismic monitoring, particularly in applications requiring detailed spatial and temporal analysis, such as structural health monitoring and environmental seismology [9].

Early DAS applications focused on exploration geophysics, particularly in borehole installations, where DAS demonstrated its potential for detailed seismic wavefield sampling [10]. Recent advancements have expanded its use to geothermal fields [11], glacier dynamics [12], and urban environments [13]. Additionally, DAS has proven effective in earthquake monitoring over various distances, from regional to tele-seismic scales [14]. This paper focuses on the comparison between DAS and the standard three-component (3C) geophones in terms of the signal quality from shot gathers and amplitude spectra in difficult conditions in the vicinity of noisy water reservoirs. Furthermore, we evaluate the performance of different DAS cable configurations and the DAS response to different seismic sources, such as sledgehammers and industrial S-wave sources. The main goal is to demonstrate the advantage of DAS in near-surface seismic applications, validating its potential as a robust extension of traditional methods. The seismic data from DAS facilitate advanced analyses such as seismic tomography and full waveform inversion (FWI), which are essential for high-resolution subsurface imaging.

## 2. Study Area and Experimental Setup

In October 2023, an active seismic survey was conducted, incorporating both DAS and three-component geophones along the same profile at the Orzepowice embankment in Poland (Figure 1). The DAS system utilized a Neubrex interrogator (NBX-S4100, a Time-Gated Digital Optical Frequency Domain Reflectometer) equipped with two types of sensors: Epsilon sensor and standard telecom cables. Additionally, standalone Digos DATA-CUBE seismic stations with 3C 4.5 Hz geophones were deployed.

A sledgehammer (5 kg, striking vertically) and an industrial S-wave source (200 kg, striking at a tilted angle) were used for wave excitation, as illustrated in Figure 2. The seismic stations were spaced at 4 m intervals, with sledgehammer shots conducted every 2 m and industrial source shots every 4 m. During the acquisition, GPS clock synchronization ensured that both the DAS and geophones recorded data simultaneously. The system recorded an initial timestamp from the GPS at the start of the measurement, after which the DAS relied on its internal clock for timing. With an accuracy of $10^{-9}$ s, the DAS internal clock is significantly more precise than the GPS timestamp, effectively eliminating the risk of time drift. Cutting out the shots was performed using timestamps from the seismic source and the DAS measurement files, following standard procedure.

## 3. DAS Configuration

DAS leverages Rayleigh backscattering from laser signals transmitted through an optical fiber using optical time domain reflectometry (OTDR). A coherent laser pulse is injected into the fiber, and backscattered waves return to the excitation point, where local fiber deformation introduces an optical phase shift between the leading and trailing parts of the pulse [15]. This phase shift alters the interference pattern, leading to variations in the intensity and phase of the returned light, which are analyzed by DAS systems as shown in Figure 2. The phase difference between the returned pulses separated by a gauge length g is controlled by the change in the optical path length. For a segment about a specific channel at position *L*, the time difference between the returned pulses is(1)$$T(L,t)=2\int_{L-g/2}^{L+g/2}dx\frac{1}{v(x,t)}=\frac{2}{C_{0}}\int_{L-g/2}^{L+g/2}dx\;n(x,t)$$
where v is the speed of light in the fiber segment, $C_{0}$ is the speed of light in vacuum, and n is the refractive index. In the presence of a seismic signal, there will be a change in the time difference arising from the combination of the change in the length of the gauge segment and the modification of the speed of light in the fiber in the presence of strain [16].

By measuring the phase differences between backscattered pulses using an interferometric process, DAS systems produce phase results over a defined “gauge length” rather than at specific points. These phase differences correlate with changes in the optical path length due to local strain [17]. The phase is then unwrapped into a continuous function, enabling strain rate measurements. The phase unwrapping limit is defined by the following formula:(2)$$\frac{∆\varphi_{max}}{f_{s}}<\pi \to ∆\varphi_{max}<\frac{f_{s}}{2f_{e}}$$
where $∆\varphi_{max}$ is the relative phase angle shift, $f_{s}$ is the sampling frequency, and $f_{e}$ is the event frequency. Equation (2) illustrates the relationship that determines the maximal phase angle shift per cycle. From this equation, we can conclude that the higher the sampling frequency, the greater the dynamic measurement accuracy [18]. In this research, we chose a sampling frequency of 5 kHz, which is significantly higher compared to the dominant seismic frequency of 20–40 Hz.

Multicomponent seismic recording, which uses both vertical and horizontal component geophones, captures the seismic wavefield more comprehensively than traditional single-element methods (Figure 3). Compressional (P-) waves, the fastest elastic waves, exhibit high signal-to-noise ratios, nearly rectilinear particle motion, and are easily generated by various sources, propagating through solid and fluid environments [19]. In contrast, S-waves have substantially lower velocities, propagate only in elastic media, and are strongly dependent on fluids or gases. The phenomenon of shear wave splitting enables anisotropy analysis using P-SV and P-SH wavefields, making converted wave data valuable for studying lithofacies changes, lithology, and amplitude anomalies on P-wave sections. It leads to the conclusion that P-waves provide clear structural images as they are not mixed with any converted waves. However, S-waves are highly sensitive to voids, cracks, and water content due to their much lower speed and wavelength, giving a better spatial resolution of the near-surface structures [20].

The DAS interrogator unit sends laser pulses down the fiber at regular intervals, with the simplest configuration ensuring scattered light from the far end is received before the next pulse. Longer pulse widths improve the signal-to-noise ratio by injecting more light but must be shorter than half the gauge length’s passage time to avoid overlapping returns [21]. The choice of gauge length significantly impacts the DAS response, requiring a balance between reducing random noise and minimizing signal distortion. The gauge length acts as a moving-average filter along the fiber, suppressing random noise but introducing notches in the response at frequencies that are multiples of the inverse wave passage time. Longer gauge lengths poorly capture low-frequency waves due to the minimal variation across the gauge, and localized strong effects influence channels over distances comparable to the gauge length [22].

The application of Time-Gated Digital Optical Frequency Domain Reflectometry (OFDR) in DAS means that the spatial resolution is dependent on the frequency range rather than the pulse duration. In general, the phase shift of the backscattered light is linearly proportional to the vibration amplitude. However, there can be found a lot of randomly distributed points along the fiber where the intensity of backscattered light is extremely low. The use of a chirp-pulse signal divided into specific frequency bands eliminates the problem of signal fading. Such division ensures that the spots of the fiber where the backscatter intensity is low does not influence the rest of the signal. The application of the chirp waveform signal enhances the SNR and eliminates occurrences of singularities during the synthesis of the component of phase shift [23]. See Table 1.

Fiber optic cables for data transmission consist of thin optical fibers with a dielectric core, approximately 10 µm in diameter for single-mode transmission. The core, surrounded by lower-refractive-index cladding and a protective coating, acts as an optical waveguide, trapping light through total internal reflection at the core–cladding interface. With a typical core refractive index of 1.4475, light propagates at approximately 69% of its speed in a vacuum. Laser sources, commonly operating at near-infrared wavelengths (~1550 nm), transmit signals via multiple pulses [24]. Special-purpose cables may contain a few single-mode fibers with protective sheaths, while commercial cables can bundle up to 100 fibers. Marine cables are designed to withstand harsh environments and offer enhanced durability. Optical fiber sensors leverage the interaction of laser light with fiber imperfections; the backscattered light provides spatial information and allows for measuring the strain rate and temperature in response to environmental changes [25].

In telecommunications, multiple protective layers shield optical fibers from mechanical damage and harsh environmental conditions. These layers enhance the durability and performance in challenging environments but may hinder the transfer of structural strain to the glass core, potentially compromising the measurement accuracy. Interlayer slippage poses a particular challenge, potentially distorting measurements or isolating the fiber from the monitored phenomena entirely. This necessitates the careful interpretation of results when using layered cables for Distributed Fiber Optic Sensing [26]. See Figure 4.

An alternative to layered cables is monolithic composite fiber optic sensors, designed specifically for civil engineering and geotechnical applications. These sensors feature a homogeneous cross-section, fully integrating the optical fiber into the composite core. This design eliminates interlayer slippage and provides the highest measurement accuracy [27]. See Table 2.

The choice of fiber, cable, and sensor is critical for the efficiency and reliability of systems utilizing DFOS technology. While a data logger is replaceable, sensors embedded in structures or the ground are not easily accessible or replaceable. Thus, ensuring their reliability and longevity is essential for maintaining the overall functionality of the system [28].

## 4. Data Processing and Analysis

Four loops of fiber optic cables were deployed in a shallow 50 cm trench along the entire 280 m seismic line, as shown in Figure 5. The results of the measurements present no difference between the sensing cable and sensor in the surface seismic survey application. However, dedicated sensors have a proven record of successfully measuring low-frequency events like, for example, landslides. Despite sensors not being a competitive solution for bare surface seismic surveys, they are still an excellent choice for permanent time-lapse monitoring systems. The proven strain transfer advantage of sensors will provide enhanced-quality data on low-frequency events like hidden landslides. For telecom cables, such small events may be missed due to the slippage between the layers.

DAS recording relies on exploiting the changes in the optical path length over a gauge length around a nominal recording point. The relative change in the optical path length along this gauge length due to disturbances passing across the fiber is dominantly controlled by the axial strain rate $e_{d}$ within the gauge. This strain rate is just the spatial derivative of the ground velocity along the fiber $v_{d}$ with respect to the distance along the fiber [29]. Under the assumption of uniform sampling along the gauge length g, the effect of averaging the axial strain rate around the reference point takes the form of Equation (3):(3)$${\dot{e}}_{d}=\frac{1}{g}\int_{-g/2}^{g/2}ds\frac{\partial \;v_{d}(s)}{\partial s}=\frac{1}{g}{\left[v_{d}\left(s\right)\right]}_{s=-g/2}^{g/2}=\frac{1}{g}[v_{d}\left(\frac{g}{2}\right)-v_{d}\left(\frac{-g}{2}\right)]$$

The averaged strain rate can be obtained by differencing the ground velocity resolved along the cable at the ends of the gauge length [16].

The raw output of Distributed Acoustic Sensing (DAS) measurements consists of an array of strain rates, expressed as the rate of strain change [µε/s]. Converting this strain rate data into seismic velocity requires a systematic process. The strain rate, which is the time derivative of strain, reflects the relative displacement changes over a given distance. To begin, the strain rate is integrated over time to calculate the strain. This strain is then processed using a low-cut filter at 4 Hz to remove the amplification of low-frequency noise [30]. Next, the filtered strain is multiplied by the gauge length to determine the displacement. Finally, the seismic velocity is obtained by differentiating the displacement with respect to time. These steps transform raw DAS strain rate measurements into seismic velocity data for further analysis, as illustrated in Figure 6 and Figure 7.

Figure 6 shows a shot gather with balanced amplitudes, illustrating the transformation of raw DAS strain rate measurements into seismic velocity data, comparable to those typically recorded by standard geophones. The untransformed raw data contain noise that obscures deeper reflections. After transforming the data to the strain domain, a 4 Hz low-cut filter is applied to remove artifacts introduced during the transformation. The signal is then converted back into the velocity domain. This processing significantly enhances the data, revealing deeper reflections and reducing noise, which leads to a clearer seismic velocity analysis.

Figure 7 focuses on a single near-offset trace, highlighted in red in Figure 6, which captures seismic data close to the source. The raw trace is noisy and after applying a 4 Hz low-cut filter to the strain data and further transforming it into velocity through differentiation, the trace reveals enhanced deeper reflections with reduced noise and sharper signals. This improvement facilitates the better identification of subsurface features, which is crucial for accurate velocity analysis and subsurface characterization.

Throughout the conversion process from strain rate to seismic velocity, error control techniques are essential for maintaining the accuracy and integrity of the final seismic velocity data. Proper spatial and temporal resolution is critical to minimize errors caused by under-sampling or over-sampling. An inadequate resolution can lead to aliasing, where high-frequency signals are incorrectly represented, or the underrepresentation of important details in the data. The calibration of the system is equally important to ensure that the measurements are accurate. This includes ensuring the correct scaling of the strain rate and displacement, as well as compensating for any sensor- or system-specific biases. Signal filtering, using a 4 Hz low-cut filter, plays a crucial role in reducing noise and artifacts that can distort the data. By applying a filter that is carefully chosen for the frequency characteristics of the seismic signals, errors introduced by low-frequency noise are minimized, ensuring that only relevant seismic information is preserved. These error control techniques, when combined, ensure that the derived seismic velocity data are reliable and suitable for accurate subsurface analysis, improving the overall quality of geophysical assessments.

## 5. Results and Discussion

Figure 8 presents a shot gather comparison between the DAS system and two horizontal components of geophones, using both a sledgehammer and an industrial S-wave source. The comparison reveals that DAS outperforms geophones in terms of the signal-to-noise ratio (SNR). This is largely due to the tenfold increase in the number of channels provided by the DAS system, which allows for much clearer interpolation and a more accurate visualization of the recorded waves. The seismic signals in the DAS data are continuous and exhibit distinct wavelet shapes, demonstrating the system’s ability to capture seismic signals with high fidelity. Furthermore, surface waves, visible at times greater than 500 ms, are clearer, more continuous, and show lower frequencies in the DAS data compared to geophones. This improved signal quality in the low-frequency range enhances the precision of surface wave analysis, such as multichannel analysis of surface waves (MASW), making DAS particularly useful for seismic monitoring applications that require detailed surface wave dispersion analysis.

While standard geophones have a better amplitude response in the seismic dominant frequency range of 20–60 Hz, DAS still performs well within this range, despite exhibiting less flatness in the amplitude spectrum. The frequency content captured by DAS, especially in the lower frequencies, is crucial for high-resolution seismic imaging techniques such as full waveform inversion (FWI), which relies on the precise registration of low-frequency data to accurately model subsurface properties. This ability to capture low-frequency content opens new possibilities for imaging complex subsurface structures in both shallow and deep Earth investigations.

In terms of spatial resolution, DAS provides a significant advantage with its centimeter spatial resolution, compared to the 2 or 5 m ones typical of standard geophone deployments. This tenfold improvement in spatial resolution allows for much finer detail in seismic data analysis, enabling more accurate travel-time tomography and seismic imaging. Even with the challenge of weaker coupling and the need for additional processing steps compared to geophones, DAS still offers a distinct advantage particularly in near-surface applications, where high resolution is essential.

Additionally, DAS adapts well to complex geological environments. Its ability to operate in settings with high seismic noise, rough terrain, and rapidly changing surface conditions makes it an excellent tool for long-term seismic monitoring. Furthermore, DAS can be easily integrated with other seismic data sources, such as geophones or accelerometers, to enhance data robustness and interpretation.

DAS demonstrates its ability to record high-amplitude signals even at near offsets, as observed with both the sledgehammer and industrial S-wave sources. The industrial source, generating a horizontal component in the shots, induces a phase change at zero offset (>100 ms), compared to the sledgehammer source, which produces vertical excitation and smooth reflection phases. This near-offset effect, which DAS captures clearly, is challenging for traditional geophones with limited resolution. DAS’s ability to capture these subtle waveforms in complex geological conditions positions it as a superior tool for imaging S-waves and capturing P-S conversions, often missed by geophones due to their angular sensitivity limitations.

The conversion algorithms for DAS data, which translate strain rate measurements into seismic velocity, are designed to minimize errors. These algorithms ensure that DAS-derived velocity models maintain a high fidelity, even in large-scale monitoring projects.

The comparison between DAS and geophones under different geological conditions (such as near-offset and complex geological settings) highlights DAS’s clear advantages, including its ability to record stronger amplitudes near offsets, capturing subtle waveforms and near-surface effects that traditional geophones might miss. This makes DAS ideal for monitoring challenging environments, from urban areas to remote landscapes.

Figure 5 highlights the differences between various cable configurations and their impact on data quality. Cables 1 and 2 were positioned on the right side of the water channel, while Cables 3 and 4 are on the left. A significant difference in reflectivity is visible in the cables (3–4) at times between 350 and 400 ms, where strong reflections are clearly detected. However, these reflections are not visible in the right-side inline cables (1–2), which suggests that the reflection is strongly directional and influenced by the position of the shots. The reflectivity differences are most pronounced at near offsets, particularly between −50 m to 50 m and at times greater than 200 ms. Those reflections are related to the SH energy component of the industrial source recorded at cables on the other side of the channel, and are missing for the inline cables.

Another notable advantage of DAS is the stability and consistency of its amplitude spectrum across different shots. Unlike geophones, which may exhibit variability between shots, the amplitude spectrum of DAS data remains more stable and reliable, ensuring better consistency in seismic data processing and interpretation. This robustness makes DAS particularly valuable for time-lapse monitoring applications, where repeatability and a high sampling density are essential for tracking changes in the subsurface over time.

As an example of recorded data application, we utilized one of the most popular geotechnical methods, the MASW technique [31]. This method, based on the dispersion of surface waves, allows for the determination of the shear wave velocity (Vs), which correlates with the soil stiffness. A key challenge is separating different wave types and accurately reconstructing the dispersion curve, particularly in noisy environments. Low-frequency waves of natural origin (1–3 Hz) can complicate analysis significantly.

In the presented study near a water reservoir, where a vertical source was used, it was challenging to distinguish between Love and Rayleigh waves (see Figure 9). The primary source of noise was waves in the reservoir, compounded by surface undulations of the water. This noise is particularly evident in the geophone recordings across the entire frequency and velocity range (0–400 m/s) (Figure 9A), rendering the MASW method impractical in such conditions.

By contrast, DAS provided a much better reconstruction of the phenomena in the 2–10 Hz frequency range, up to 400 m/s velocity. DAS offered a clean and unambiguous representation of the dispersion curve, especially in the key frequency and velocity range, crucial for obtaining a reliable 1D model of near-surface ground structures. This allowed for the effective separation of individual curves.

Using DAS data, we reconstructed a 1D ground model to a reliable depth of 13 m, with an RMS error of 3.26% using the genetic algorithm from the WinMASW 2023 package, compared to 6.2% for the geophone data. After converting the strain data into velocity units, the MASW results for DAS and geophones are expressed in the same velocity units (m/s), ensuring consistency in the analysis. Based on the 1D results (Figure 9C), we modeled the fundamental modes for Love and Rayleigh waves (Figure 9D).

A notable advantage of DAS is its ability to record a wider frequency range compared to the 4.5 Hz geophones, enabling better wave-type separation and improved ground model quality. This is evident in subsection D, where the synthetic Love wave fits well with the real dispersion curve, even though Love waves were not directly used for the MASW calculations. The MASW data processing was also optimized to improve wave-type separation and velocity inversion, ensuring high accuracy in the final ground model.

## 6. Conclusions

DAS technology presents an interesting possibility to enhance traditional seismic surveys in near-surface studies. Compared to standard geophones, DAS technology requires one additional processing step to transfer the recorded wavefield data to the typical velocity domain. The sources in both cases are the same and no additional efforts are needed. The problem of coupling the sensor to the ground, often mentioned in the literature, is fully neglected when digging cables in shallow trenches. The quality of the recorded data is comparable, but still, DAS technology records near-field energies in a cleaner way and works precisely even close to strong industrial sources. For the described near-surface applications in the Orzepowice embarkment, we observe no differences for the data recorded using two types of fiber sensors. DAS provides a tenfold higher spatial resolution and wider frequency range, including maximal frequencies three orders of magnitude higher than those from the geophone acquisition. These characteristics favor the use of fiber optic technology for near-surface imaging with S-waves. The presented results demonstrate this potential in an easier and more precise analysis using the MASW technique. Finally, DAS seems to be an excellent solution for the permanent time-lapse monitoring of critical infrastructure objects, as the survey geometry is perfectly maintained and the sensors themselves are protected from natural elements and vandalism. Additionally, the use of fiber optic sensors may extend the system monitoring capabilities by enhancing the sensing of low-frequency events [32]. This solution would create very robust and reliable sensing network, ensuring safety and reliability. As a negative factor, it is necessary to mention the large volumes (terabytes) of raw data recorded during the survey, the fragility of the system in the field, and high cost of the interrogators. We hope that this technology will develop in the near future to be even more useful in near-surface applications.

## Figures and Tables

**Figure 1 sensors-25-01558-f001:**
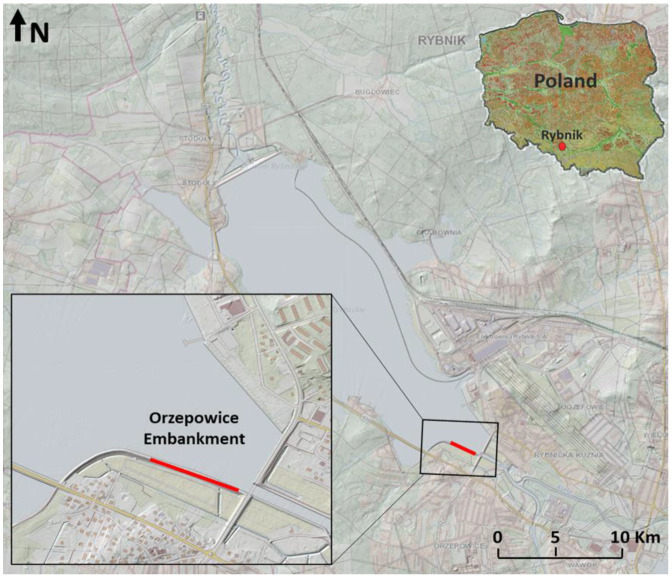
Base map for the study area on the Rybnik reservoir. The red line marks the profile localization on the embankment.

**Figure 2 sensors-25-01558-f002:**
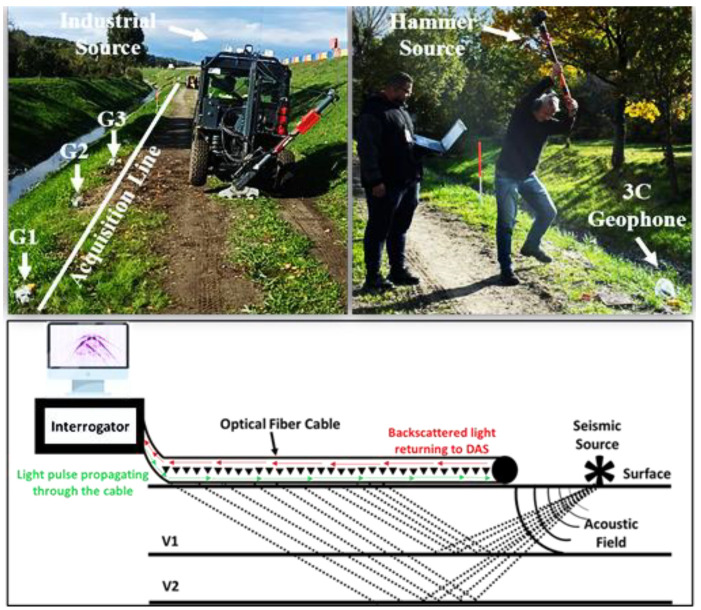
Configuration showing the seismic acquisition line, both seismic sources, and the DAS basic concept.

**Figure 3 sensors-25-01558-f003:**
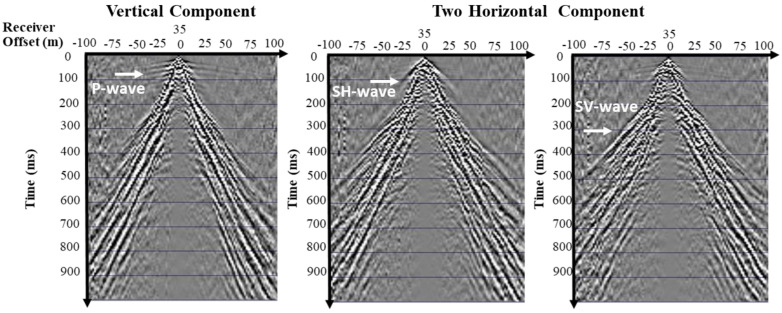
Vertical-component geophone using hammer source to show P-waves and two horizontal-component geophone using hammer source to show S-wave (SH and SV) in the receiver domain.

**Figure 4 sensors-25-01558-f004:**
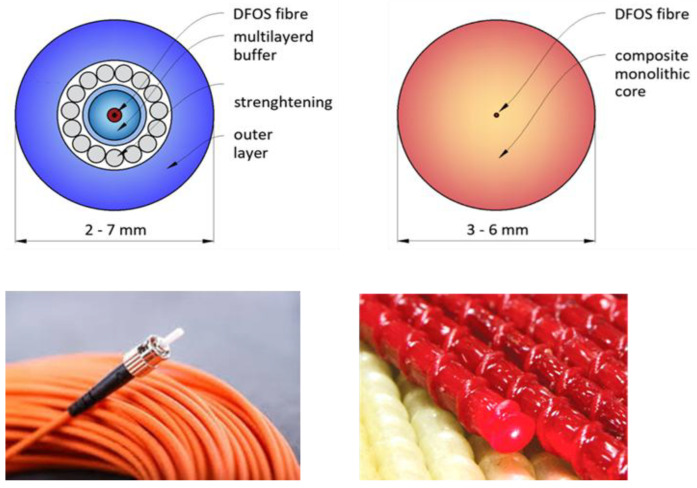
Comparison between layered cables and monolithic sensors.

**Figure 5 sensors-25-01558-f005:**
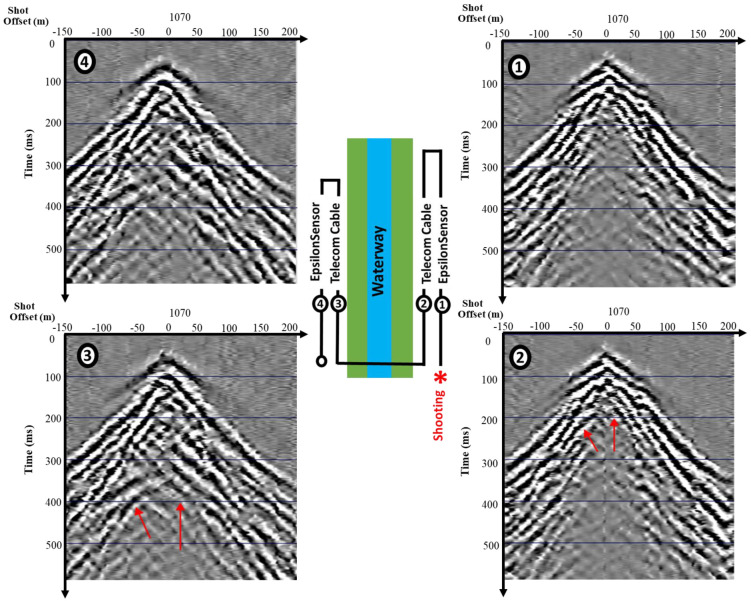
Comparison between Epsilon sensor and telecom cables in the shot domain. Middle subplot shows cable geometry on both sides of the water channel.

**Figure 6 sensors-25-01558-f006:**
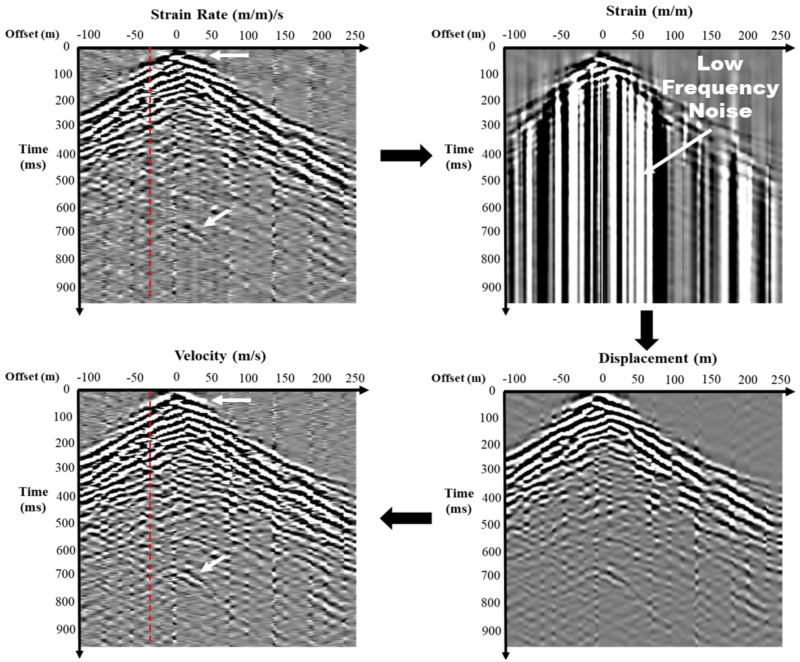
Shot gather illustrating the transformation of raw DAS strain rate measurements into seismic velocity data for analysis.

**Figure 7 sensors-25-01558-f007:**
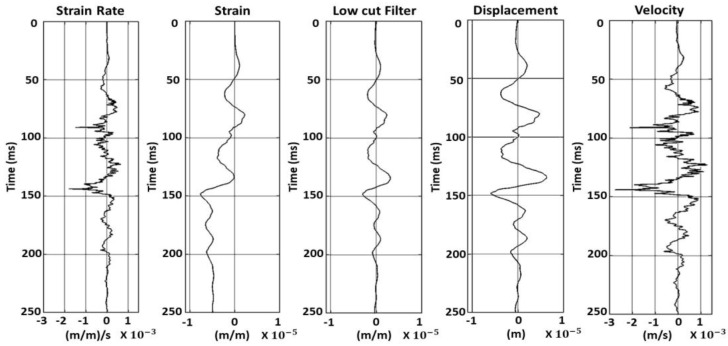
Near-offset trace illustrating the transformation of raw DAS strain rate measurements into seismic velocity data for analysis.

**Figure 8 sensors-25-01558-f008:**
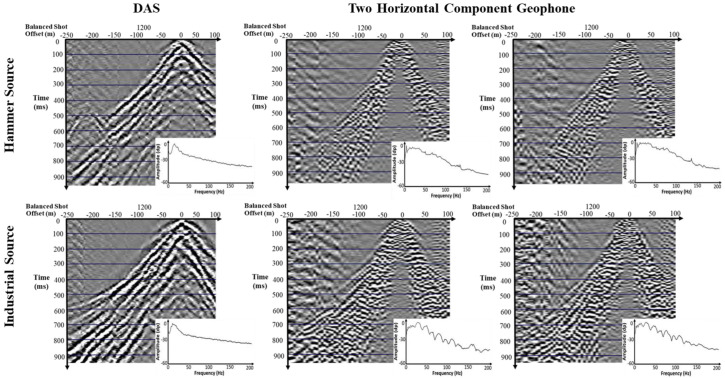
Shot gather comparison: DAS vs. two horizontal geophone components using sledgehammer and industrial S-wave sources.

**Figure 9 sensors-25-01558-f009:**
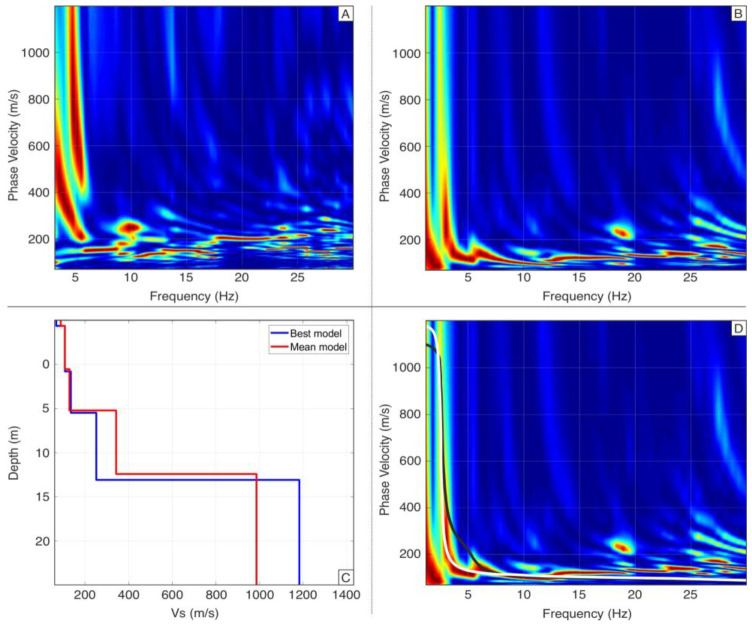
Comparison of dispersion curves obtained from geophones (**A**) and fiber optic acquisition (**B**) for the same shot and horizontal “in-line” component. The MASW result (**C**) and theoretical dispersion curves (**D**) for Love waves (white line) and Rayleigh waves (black line) show a good fit to the recorded data.

**Table 1 sensors-25-01558-t001:** Field parameters for Distributed Acoustic Sensing acquisition.

Device model	Neubrex NBX-S4100
Distance range	1100 [m]
Used fiber	Epsilon sensor and telecom cable
Gauge length	2 m
Spatial resolution	20 cm
Spatial sampling	20 cm
Sampling frequency	5000 Hz
Chirp pulse power	18 dBm

**Table 2 sensors-25-01558-t002:** Comparison between Epsilon sensor and telecom cables.

Product	EpsilonSensor	Telecom Cable
Type	Monolithic	Layered
Surface	Ribbed	Smooth
Diameter	5mm	5.3mm
Operating temperature	−20 to +80 °C	−20 to +50 °C
Material	PLFRP (polyester fiber + epoxide)	PE
Elastic modulus	3GPa	Unknown
Strain transfer method	Direct	Indirect

## Data Availability

The data will be available in the data portal of the Institute of Geophysics, Polish Academy of Sciences (https://dataportal.igf.edu.pl/dataset?organization=geophysical-imaging), after the finalization of the project in 2027. During the analysis, the commercial software Globe Claritas, developed by PetroSys, was used and geodetic conversions were performed in QGIS software.

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
