# Peer review of "Application of Distributed Acoustic Sensing for Active Near-Surface Seismic Monitoring"

_sensors, 2025, doi:10.3390/s25051558_

Round 1

Reviewer 1 Report

Comments and Suggestions for Authors

The paper investigates the application of Distributed Acoustic Sensing (DAS) technology in near-surface seismic monitoring and compares it with traditional 3C geophones. However, after a thorough review, I believe there are several significant issues that prevent the paper from meeting the publication requirements. The main concerns are as follows:

 1. Lack of Novelty

While DAS technology is widely used in seismic monitoring, the manuscript fails to present sufficient novelty. Many previous studies have already explored the use of DAS in seismic monitoring and compared it with traditional geophones. The experimental setup, analysis methods, and conclusions presented in this paper are highly similar to existing literature, lacking a significant new contribution.

 2. Insufficient Depth in Data Analysis

Although the manuscript compares DAS and 3C geophones in terms of signal quality, the analysis is superficial. The paper mainly focuses on signal-to-noise ratio (SNR), amplitude spectra, and surface wave analysis, but it does not delve deeper into the adaptability of DAS data in complex geological environments, its long-term monitoring capabilities, or the feasibility of integrating DAS with other seismic data. Furthermore, the conversion algorithms for DAS data (converting strain rate to seismic velocity) are not adequately explained in terms of error control and reliability, which diminishes the credibility of the conclusions.

 3. Limitations in Experimental Design

The experiments were conducted only in a water reservoir embankment area, using a single seismic source (sledgehammer and industrial S-wave source). The geological environment is relatively simple, and the study lacks a broader applicability. The manuscript does not cover the performance comparison of DAS and geophones in different geological conditions or explore the feasibility of DAS in larger-scale real-world applications (such as urban seismic monitoring or large-scale earthquake early warning systems).

Comments on the Quality of English Language

non

Author Response

The paper investigates the application of Distributed Acoustic Sensing (DAS) technology in near-surface seismic monitoring and compares it with traditional 3C geophones. However, after a thorough review, I believe there are several significant issues that prevent the paper from meeting the publication requirements. The main concerns are as follows:

Thank you for your in-dept review. You understood the subject well and you suggest changes that really improve our manuscript.

  1. Lack of Novelty

While DAS technology is widely used in seismic monitoring, the manuscript fails to present sufficient novelty. Many previous studies have already explored the use of DAS in seismic monitoring and compared it with traditional geophones. The experimental setup, analysis methods, and conclusions presented in this paper are highly similar to existing literature, lacking a significant new contribution.

It is true that this manuscript is not ground breaking in the field, but we think it as a solid comparison of innovative and standard sensors types. Yes similar studies were conducted, but we found no article related to P and S waves observations for various source types in levees/dams studies. Often such results are not being published but only presented in engineering conferences. We are convinced that publishing these results in the Sensors will broaden the range of readers involved in geohazard problem and allows them to understand this new technology.

  1. Insufficient Depth in Data Analysis

Although the manuscript compares DAS and 3C geophones in terms of signal quality, the analysis is superficial. The paper mainly focuses on signal-to-noise ratio (SNR), amplitude spectra, and surface wave analysis, but it does not delve deeper into the adaptability of DAS data in complex geological environments, its long-term monitoring capabilities, or the feasibility of integrating DAS with other seismic data. Furthermore, the conversion algorithms for DAS data (converting strain rate to seismic velocity) are not adequately explained in terms of error control and reliability, which diminishes the credibility of the conclusions.

In the manuscript prepared for Sensor journal we tried to focus on sensors related observations and not dive deeply to geophysical processing. That is why we present results focused on SNR and observed amplitude spectra, but we used a simple MASW example to show the advantage of combining DAS with standard geophones. We agree with your comment that DAS data conversion should be described in more details. To fix this we added description in lines 260-273 in Data processing chapter, and in lines 293-295, 303-307, 320-328, 369-371 in the Discussion chapter.

  1. Limitations in Experimental Design

The experiments were conducted only in a water reservoir embankment area, using a single seismic source (sledgehammer and industrial S-wave source). The geological environment is relatively simple, and the study lacks a broader applicability. The manuscript does not cover the performance comparison of DAS and geophones in different geological conditions or explore the feasibility of DAS in larger-scale real-world applications (such as urban seismic monitoring or large-scale earthquake early warning systems).

Indeed our survey was planned in embankment area with a simple geological structure. The reason was to compare both sensor types and both sources in relatively simple geology so we could focus on more general problems like ambient noise or sensors response. In complex geology such sensors related characteristics would be superimposed on geological complexity in the wavefield and most probably would not be possible to observe. Moreover, our manuscript was focused on Sensors journal, thus we put our efforts to sensors related observations. In the future we will present more data for various geological complexity and more advanced seismic processing, but that will be aimed for Geophysical focused journals.

Reviewer 2 Report

Comments and Suggestions for Authors

The manuscript "Application of Distributed Acoustic Sensing for Active Near-Surface Seismic Monitoring" presents a the application of a method (DAS) that attracting great interest last years in a prototype study including near surface monitoring. The authors prepares a well structured manuscript, providing sufficient information for materials and methods used. The presented findings are supported adequately from the results. 

I have only one major concern regarding the application of MASW method regarding the data used for MASW calculations. If the data came from DAS units then i suggest the authors to perform new measurements using geophone configurations in order to have compatible results for comparison. Otherwise, if the data came from geophone measurements then the authors must provide more detail about the configuration and the data processing. Leaving only results MASW from DAS unit does not help the readers to understand the superiority of your method. You must provide also MASW results from geophones and compare MASW results from the different measurement instruments (DAS and 3C seismic stations)

Author Response

The manuscript "Application of Distributed Acoustic Sensing for Active Near-Surface Seismic Monitoring" presents a the application of a method (DAS) that attracting great interest last years in a prototype study including near surface monitoring. The authors prepares a well structured manuscript, providing sufficient information for materials and methods used. The presented findings are supported adequately from the results. 

I have only one major concern regarding the application of MASW method regarding the data used for MASW calculations. If the data came from DAS units then i suggest the authors to perform new measurements using geophone configurations in order to have compatible results for comparison. Otherwise, if the data came from geophone measurements then the authors must provide more detail about the configuration and the data processing. Leaving only results MASW from DAS unit does not help the readers to understand the superiority of your method. You must provide also MASW results from geophones and compare MASW results from the different measurement instruments (DAS and 3C seismic stations)

Thank you for such positive opinion about the manuscript.

In fact presented MASW analysis was performed for both geophone and DAS data, as you are suggesting. As presented in Figure 9 we compare the results of both: geophones data in panel A and DAS data in panel B. To make it clearer we updated the text in lines 320-328 and figure 9 figure caption to underline this.

Reviewer 3 Report

Comments and Suggestions for Authors

The MS uncder review presents a carefully made research, from an informative introduction through the sections presenting the experimental circumstances and equipments to the discussion section and conclusions. The description of the techniques used, the methods of data acquiring and their interpretation, and the conclusions are relevant, self-consistent and novel. The English grammar of the MS is nearly excellent. Thus, given all said above, I strongly recommend this MS for publication in "Sensors" as it is now. 

Author Response

The MS uncder review presents a carefully made research, from an informative introduction through the sections presenting the experimental circumstances and equipments to the discussion section and conclusions. The description of the techniques used, the methods of data acquiring and their interpretation, and the conclusions are relevant, self-consistent and novel. The English grammar of the MS is nearly excellent. Thus, given all said above, I strongly recommend this MS for publication in "Sensors" as it is now. 

Thank you for such positive feedback and your strong recommendation for publication. As you suggested no modifications we could not use them, but we modified the manuscript according to suggestions of other reviewers. I hope they are acceptable, as they are not modifying the main scope of the manuscript.

Round 2

Reviewer 1 Report

Comments and Suggestions for Authors

Fig. 3, 5,6,8,9, all lack Z-axis coordinate information.

Author Response

Fig. 3, 5,6,8,9, all lack Z-axis coordinate information.

Dear Reviewer thank you for such positive comment. We understand that you like our manuscript and have no suggestions to the text. Yes, we agree with your suggestions about the vertical scale annotation if figures 3, 5 and 8, and we correct this in updated manuscript. However, we found annotations in figures 6 and 9 to be correct, thus we are not modifying them.